# IS EXTENDING MODALITY THE RIGHT PATH TOWARDS OMNI-MODALITY?

## ABSTRACT

Omni-modal language models (OLMs) aim to integrate and reason over diverse input modalities—such as text, images, video, and audio—while maintaining strong language capabilities. Despite recent advancements, existing models, especially open-source ones, remain far from true omni-modality, struggling to generalize beyond the specific modality pairs they are trained on or to achieve strong performance when processing multi-modal inputs. We study the effect of extending modality, the dominant technique for training multimodal models, where an off-the-shelf language model is fine-tuned on target-domain and language data. Specifically, we investigate three key questions: (1) Does modality extension compromise core language abilities? (2) Can model merging effectively integrate independently fine-tuned modality-specific models to achieve omni-modality? (3) Does omni-modality extension lead to better knowledge sharing and generalization compared to sequential extension? Through extensive experiments, we analyze these trade-offs and provide insights into the feasibility of achieving true omni-modality using current approaches.

## 1 INTRODUCTION

Omni-modal language models (OLMs) refer to models that can accept and understand various input modalities—text, images, video, audio, etc.—and engage with users with language in a seamless, natural manner. Ideal OLMs are able to combine inputs from different modalities into a unified perception of real-world scenarios, enabling deeper contextual comprehension and more comprehensive reasoning. This capability would empower embodied (Ma et al., 2024) and virtual (Deng et al., 2023) agents to perceive their environment.

OLMs belong to the broader category of multimodal models. While recent advancements have made substantial progress in this field (OpenAI, 2024; Li et al., 2024), current models still lack true omni-modality—the ability to handle arbitrary modality combinations while maintaining robust reasoning and interaction abilities, evidenced by their inability to generalize beyond the specific modality pairs they were trained on. For example, models trained on text-image tasks (Liu et al., 2023a; Li et al., 2023) struggle with video understanding, and models optimized for text-video tasks (Fu et al., 2024; Zhang et al., 2024c) often fail to incorporate spatial reasoning from static images. Additionally, open-source OLMs (Wu et al., 2024b) following often exhibit weaker performance on benchmarks specifically designed for fewer modality evaluation, such as text-image tasks, compared to modality-specific models.

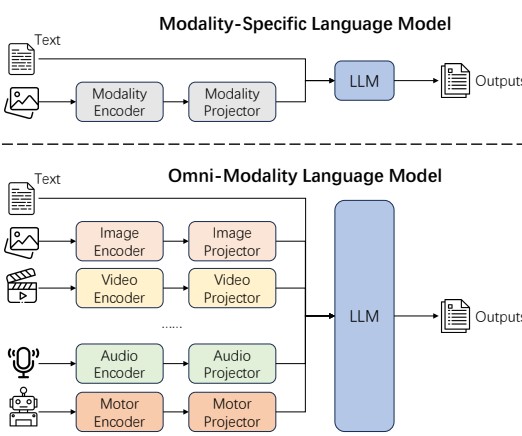

Figure 1: Overview of modality-specific language models and Omni-modal language models.

Notably, almost all existing multimodal models rely on a common strategy: *extending modality*. As shown in Figure 1, this technique fine-tunes an off-the-shelf large language model (LLM) on

data that pairs language with target modalities, enabling rapid adaptation to multimodal tasks (Li et al., 2024; Wu et al., 2024b). However, the extent to which extending modality contributes to the limitations of current multimodal models remains unclear (e.g., its impact on fundamental language capabilities). Since LLMs serve as the backbone for most multimodal models, an important question is *RQ1: whether modality extension compromises their core language abilities* in favor of a stronger performance on modality-specific benchmarks. Do models retain their original reasoning and linguistic proficiency, or does the introduction of additional modalities interfere with their generalization across language tasks?

With the abundance of modality-extended models in the open-source community, another open question is: *RQ2: Can we preserve their abilities while extending multimodal capabilities using existing models?* Model merging strategies (Wortsman et al., 2022) have been explored in various applications to combine knowledge from different models, often improving performance on downstream tasks. However, it remains unclear whether such techniques can be leveraged to create an effective OLM by merging multiple modality-extended models trained on different modalities. Would merging independently fine-tuned models allow them to integrate cross-modal knowledge effectively, or would inconsistencies between separate modality-extended models hinder the fusion process?

Furthermore, *RQ3: Does omni-modality fine-tuning lead to a more effective OLM?* While some multimodal models have been trained to handle multiple modalities simultaneously, there has been no systematic comparison of different strategies for extending multiple modalities. Most current approaches add one modality at a time through sequential fine-tuning (Zhang et al., 2024c), yet it remains unclear whether this stepwise process is more effective than omni-modality fine-tuning. Would joint training across multiple modalities improve knowledge sharing and downstream performance, or is task-specific fine-tuning a more efficient approach?

Our findings highlight the impact of modality fine-tuning and the limitations it presents in the pursuit of OLMs. First, we identify a trade-off between extending modalities and preserving the core language capabilities. While modality fine-tuning can enhance certain LLM abilities, especially in areas like knowledge extension where visual modalities (e.g., images and videos) provide significant improvements, it tends to degrade crucial functions such as reasoning and instruction-following. Second, we introduce and compare weighted average model merging with standard average merging. Our results show that weighted model merging achieves the best performance across both textual and multimodal tasks, successfully preserving the most critical attributes of the original LLM, with parameter shifts acting as indicators of importance. We also demonstrate that each attention head in modality fine-tuned models is integral to completing modality-specific tasks. Third, we compare omni-modality fine-tuning with modality-specific fine-tuning, revealing that, while omni-modality fine-tuning holds conceptual appeal, it is less effective and efficient than models fine-tuned for specific modalities. Moreover, we experiment the small-step fine-tuning (Cohere et al., 2025) on the weighted merged model. The results show that although small-step fine-tuning works on merged language models, it fails in omni-modal models.

## 2 RELATED WORK

**Multimodal Large Language Models.** Multimodal large language models (MLLMs; (OpenAI, 2024; Zhang et al., 2024a)) have gained significant attention due to their ability to process and reason across different modalities, including text, images, video, and audio. Recent approaches (Alayrac et al., 2022; Li et al., 2022) have explored fine-tuning language models with modality-specific data to enable them to handle multimodal inputs directly. For instance, BLIP-2 (Li et al., 2023) and LLaVA (Liu et al., 2023a) focus on extending image understanding capabilities, while Video-LLaMA (Zhang et al., 2023) extends video understanding. However, these models still suffer from modality issues, such as hallucinations (Huang et al., 2025; 2024) and knowledge conflicts (Zhu et al., 2024). Furthermore, the impact of applying fine-tuning using modality-specific data on the original LLM remains unclear (Fei et al., 2025).

**Omni-Modal Language Models.** Omni-modal language models (OLMs; (Hurst et al., 2024)) aim to create a unified framework that can simultaneously process and reason about various modalities without the need of separate models for each type of input. Recent studies, such as NextGPT (Wu et al., 2024b) and OLA (Liu et al., 2025), have attempted fine-tuning with multiple modality-specific data, integrating text, image, and video understanding into a single LLM. These models leverage

shared latent spaces to enhance cross-modal understanding (Lu et al., 2022), enabling more coherent understanding across heterogeneous data. However, the effectiveness and efficiency of extending multiple modalities from the base LLM remains unclear, leaving the effective training pattern of OLMs indefinitive.

**Continual Learning.** Continual learning for large language models (Wu et al., 2024c; Yang et al., 2025) focuses on teaching LLMs from a continuous data stream over time, enabling knowledge expansion (Kowalczyk et al., 2024; Wang et al., 2025) and conflict dissolving (Du et al., 2024). Continual learning can be categorized into: continual pre-training (Qin et al., 2023; Yu & Ji, 2024), continual fine-tuning (Wu et al., 2024a; Minixhofer et al., 2024), and continual alignment (Yao et al., 2023; Zhang et al., 2024b). Recent study has revealed several drawbacks of continual learning, including general performance deterioration and weakened safety (Li et al., 2025). However, the impact of extending modality, which can be viewed as continual fine-tuning, is still under-explored.

Table 1: Overview of base LLMs and their corresponding multimodal models (MLLMs), including the modalities they support and the associated modality extension data used for training.

| Base LLM | MLLM | Modality | Modality Extension Data |
|---|---|---|---|
| Qwen2-7B-Instruct | Qwen2-VL-7B-Instruct | Image & Video | >1.4 trillion mutlimodal tokens |
| | LLaVA-OneVision-Qwen2-7B-SI | Image | 8.5m image data |
| | LLaVA-Video-7B-Qwen2 | Video | 8.5m image data + 1.6m image & video data |
| Qwen-7B-Instruct | Qwen2-Audio-7B-Instruct | Audio | 520k audio instruction pairs |
| Vicuna-7B-V1.5 | LLaVA-1.5-7B | Image | 600k image insturction pairs |
| Qwen2-72B-Instruct | Qwen2-VL-72B-Instruct | Image | >1.4 trillion multimodal tokens |

# 3 PROBLEM SETUP

Before discussing how MLLMs extend modalities based on LLMs, we first formalize the key components of MLLMs and the process of modality fine-tuning.

**MLLM Architecture.** Contemporary MLLMs (Liu et al., 2023a; Li et al., 2023) adopt a general architecture which consists of a base LLM, modality encoders $\mathcal{E} = \{E_{m_1}, E_{m_2}, ..., E_{m_n}\}$, and modality projectors $\mathcal{P} = \{P_{m_1}, P_{m_2}, ..., P_{m_n}\}$, where each $m_i, i \in [1, 2, ..., n]$ is a modality except the textual one. Given a multimodal input $\mathcal{X} = \{x_{m_0}, x_{m_1}, x_{m_2}, ..., x_{m_n}\}$, where $m_0$ is the textual modality, for each modality $m_i$, the MLLM first encode the modality input $x_{m_i}$ using $E_{m_i}$. Then, a projector $P_{m_i}$ projects the encoded input to the textual modality as $t_{m_i}$. In the meantime, the language tokenizer tokenizes the textual input $x_{m_0}$ into a token sequence $t_{m_0}$. The LLM takes the encoded multimodal input $\mathcal{T} = \{t_{m_0}, t_{m_1}, t_{m_2}, ..., t_{m_n}\}$ and generates the output by the probability $p_{\text{LLM}}(y|\mathcal{T})$. As $m_i$ covers all the modalities, MLLM becomes OLM.

**Modality Fine-Tuning.** Modality fine-tuning is a common way to extend modalities on LLMs. Specifically, modality fine-tuning leverages modality-specific instruction data—including various tasks on a certain modality—to fine-tune a LLM to capture the inputs encoded by modality encoders and modality projectors. There are two popular ways to achieve this: to *freeze* the LLM and only train the modality encoder and projector or to *fine-tune* the LLM with modality instruction data. Recent studies have pointed out that unfreezing the LLM for modality fine-tuning is essential for keeping the most desirable attributes of LLMs, such as in-context learning (Lin et al., 2024). Thus, in this paper, we discuss modality fine-tuning in the context of unfreezing the LLM.

**General Experimental Setup.** For all the experiments, we adopt the default generation parameters. For multiple-choices datasets, we adopt greedy decoding to generate the options. For datasets that require sampling, we set the temperature to 1.0.

# 4 ON THE IMPACT OF MODALITY FINE-TUNING ON BASE LLM

Despite the success of extending modality on LLMs, modality fine-tuning without freezing the base LLM, which alters its default parameters, potentially affects its original performance. While some studies have discussed preserving the base LLM's capabilities (Xu et al., 2025b), the broader implications of modality fine-tuning remain largely underexplored. In this section, we examine how fine-tuning on different modalities influences the base language model.

Table 3: Performance of the base LLMs across all evaluated domains of textual abilities. We find that 1) video modality enriches the parametric knowledge. 2) Modality fine-tuning harms major core language abilities. 3 ) Video modality may enhance long context.

| Model | MMLU | MMLU-Pro | IFEval | PR | ZeroScrolls | GPQA | MATH | HumanEval+ | | | MMMLU | HarmBench |
|---|---|---|---|---|---|---|---|---|---|---|---|---|
| | Acc | Acc | Avg | Acc | Acc | Acc | Acc | Pass@1 | Pass@5 | Pass@10 | Acc | ASR@100↓ |
| **Qwen2-7B based** | | | | | | | | | | | | |
| Qwen2-7B-Instruct | 66.2 | 35.1 | 58.2 | 100.0 | 85.7 | 13.6 | 60.0 | 54.2 | 82.8 | 93.4 | 51.4 | 15.9 |
| Qwen2-VL-7B-Instruct | 66.5↑ | 40.0↑ | 48.6↓ | 100.0= | 90.5↑ | 10.6↓ | 49.8↓ | 37.6↓ | 75.4↓ | 92.1↓ | 51.4= | 11.4↑ |
| LLaVA-Video-7B-Qwen2 | 66.6↑ | 37.7↑ | 48.8↓ | 100.0= | 76.2↓ | 8.6↓ | 47.2↓ | 41.8↓ | 80.2↓ | 92.1↓ | 48.9↓ | 36.0↓ |
| LLaVA-OneVision-Qwen2-7B-SI | 65.4↓ | 37.9↑ | 28.6↓ | 100.0= | 76.2↓ | 4.6↓ | 14.6↓ | 0.0↓ | 0.0↓ | 0.0↓ | 48.2↓ | 34.1↓ |
| **Qwen-7B based** | | | | | | | | | | | | |
| Qwen-7B-Chat | 38.4 | 14.9 | 25.7 | 35.5 | 0.0 | 5.1 | 8.6 | 0.0 | 0.0 | 0.0 | 33.9 | 8.2 |
| Qwen2-Audio-7B-Instruct | 41.9↑ | 12.1↓ | 19.4↓ | 30.5↓ | 14.3↑ | 4.6↓ | 2.2↓ | 0.0= | 0.0= | 0.0= | 30.0↓ | 0.0↑ |
| **Vicuna-7B based** | | | | | | | | | | | | |
| Vicuna-7B-V1.5 | 45.5 | 17.8 | 41.8 | 50.0 | 0.0 | 10.1 | 6.1 | 8.0 | 24.3 | 43.9 | 30.1 | 24.6 |
| LLaVA-1.5-7B | 48.9↑ | 20.9↑ | 37.6↓ | 50.0= | 9.5↑ | 8.6↓ | 19.6↓ | 6.8↓ | 22.6↓ | 42.0↓ | 35.3↑ | 44.4↓ |
| **Qwen2-72B based** | | | | | | | | | | | | |
| Qwen2-72B-Instruct | 79.0 | 48.7 | 81.4 | 100.0 | 85.7 | 10.1 | 70.0 | 72.5 | 88.1 | 93.4 | 67.0 | 1.5 |
| Qwen2-VL-72B-Instruct | 81.5↑ | 50.2↑ | 62.9↓ | 100.0= | 90.5↑ | 9.6↓ | 64.7↓ | 47.7↓ | 88.0↓ | 96.7↑ | 68.7↑ | 11.2↓ |
| **Qwen3-4B based** | | | | | | | | | | | | |
| Qwen3-4B-Instruct | 69.3 | 42.8 | 89.2 | 100.0 | 47.6 | 12.6 | 71.8 | 81.3 | 94.4 | 96.0 | 56.3 | 0.0 |
| Qwen3-VL-4B-Instruct | 69.7↑ | 44.0↑ | 86.2↓ | 100.0= | 76.2↑ | 12.6= | 66.3↓ | 80.0↓ | 89.2↓ | 93.4↓ | 55.9↓ | 0.0= |

## 4.1 EXPERIMENTAL SETUP

**Datasets.** To systematically assess the impact of modality fine-tuning, we evaluate six core LLM abilities: *Knowledge*, *Instruction Following*, *Long Context*, *Reasoning*, *Multilingual*, and *Safety*. The statistics of these datasets are listed in Table 2. *1)* For *Knowledge*, we adopt MMLU (Hendrycks et al., 2020a) and MMLU-Pro (Wang et al., 2024c). *2)* For *Instruction Following*, we adopt IFEval (Zhou et al., 2023). We report an average performance across strict prompt, strict instruction, loose prompt, loose instruction. *3)* For *Long Context*, we adopt Passkey Retrieval (Mohtashami & Jaggi, 2023) and the Quality subset of ZeroScrolls (Shaham et al., 2023). *4)* For *Reasoning*, we evaluate three different domains, namely, general reasoning GPQA (Rein

Table 2: Benchmarks for textual abilities.

| Task | Dataset | Size |
|---|---|---|
| Knowledge | MMLU | 14,079 |
| | MMLU-Pro | 12,032 |
| Instruction Following | IFEval | 541 |
| Long Context | Passkey Retrieval | 400 |
| | ZeroScrolls/Quality | 21 |
| Reasoning | GPQA | 198 |
| | MATH | 5,000 |
| | HumanEval++ | 164 |
| Multilingual | MMLU | 196,588 |
| Safety | Harmbench | 200 |

et al., 2024), math reasoning MATH (Hendrycks et al., 2021), and coding HumanEval-plus (Liu et al., 2023b). *5)* For *Multilingual*, we adopt MMMLU (Hendrycks et al., 2020b). *6)* For *Safety*, we adopt HarmBench (Mazeika et al., 2024).

**Models.** Supervised fine-tuning on different modalities can steer LLMs in diverse directions. Intuitively, fine-tuning on image data may enrich the model's contextual understanding, while video fine-tuning may enhance its ability to process long-range dependencies. To systematically analyze these effects, we conduct controlled experiments across different modalities and model sizes.

Our primary analyses adopts the Qwen2-7B-Instruct model family (Yang et al., 2024), as the following multimodal extensions of Qwen2-7B-Instruct easily supports a comprehensive comparison across modalities and Table 1 shows the detailed model statistics. *1)* **Image** modality: Qwen2-VL-7B-Instruct (Wang et al., 2024b) and LLaVA-OneVision-Qwen2-7B-SI (Li et al., 2024). *2)* **Video** modality: LLaVA-Video-7B-Qwen2 (Zhang et al., 2024c). *3)* **Audio** modality: Qwen2-Audio-7B-Instruct (Chu et al., 2024). To evaluate whether these findings generalize across different base LLMs, we also test Vicuna-7B-V1.5 (Chiang et al., 2023), alongside its image extension LLaVA-1.5-7B (Liu et al., 2023a). Also, we assess the impact of model size by analyzing Qwen2-72B-Instruct (Wang et al., 2024b) and its visual extension, Qwen2-VL-72B-Instruct (Yang et al., 2024).

## 4.2 RESULTS

The evaluation results are presented in Table 3, from which we derive several key observations.

**Visual modality extends the scope of parametric knowledge, with gains scaling directly with the quantity of training data.** Results from MMLU and MMLU-Pro show that visual extension of Qwen2 improve performance by at least 2.5%. Specifically, Qwen2-VL, trained on over

1.4T multimodal tokens, achieves an approximately 5% improvement, compared to a 2.5% gain for LLaVA-Video-7B-Qwen2, trained on 10M instruction data. This supports the hypothesis that visual fine-tuning effectively injects new visual knowledge into the base model. Conversely, the audio modality provides minimal knowledge expansion, improving Qwen2's performance by only 0.4%. This disparity arises because audio primarily acts as an extension of natural language, unlike vision which introduces novel knowledge forms. This implies that fine-tuning paradigms must be modality-specific: audio should prioritize alignment with text, while vision should focus on synergizing multimodal representations to integrate new knowledge effectively.

**Modality fine-tuning harms instruction following, reasoning, and safety.** Modality fine-tuning significantly degrades instruction-following capabilities, with all extended models showing performance declines on the IFEval dataset. This suggests current fine-tuning methods act only as modality extensions and fail to preserve this core skill, indicating a need to co-train with instruction-following data to mitigate the issue (Jindal et al., 2024). Reasoning performance is also severely impacted across different domains: 3.0% on GPQA, 10.2% on MATH, and declines in both Pass@1 and Pass@5 on HumanEval+ with the best-performing model Qwen2-VL. While 7B-scale models exhibit substantial drops, this degradation is less pronounced in larger models; the 72B Qwen2-VL's accuracy drop on MATH was only 5.3%, possibly because idle parameters in larger models absorb the fine-tuning impact. Nevertheless, a significant overall decline in reasoning ability remains. On the HarmBench dataset, nearly all multimodal models exhibit a higher attack success rate, indicating reduced safety compliance. This aligns with previous findings that modality fine-tuning disrupts the model's existing RLHF alignment (Lee et al., 2025).

**Video modality may enhance the long context ability.** From the results of the ZeroScrolls dataset, we can observe that those models trained on large amounts of video data, except for those base LLMs do not have the long context ability, show an increase in the performance, *i.e.* the Qwen2-VL-7B-Instruct and Qwen2-VL-72B-Instruct. On the contrary, the LLaVA-OneVision model, which is trained on single image data, shows a decrease in the long context performance. Intuitively, video is a visual version of the long-context document. A typical video fine-tuning sample contains roughly 4k tokens (Zhang et al., 2024c), mostly visual tokens. Thus, training on video data could inherently enhance the long context understanding ability of the base LLM.

**Modality fine-tuning has a mixed effect on multilingual performance.** The results on the MMMLU dataset reveal varying effects of modality fine-tuning across different models and model sizes. In vicuna-based models, the image extension enhances multilingual performance by 5.2%, whereas in Qwen2-based models, performance either declines or remains unchanged. However, in larger models, multilingual performance improves by 1.7%.

## 5 ON MODEL MERGING TOWARDS AN OMNI-MODAL LANGUAGE MODEL

Having gained a clearer insight into the impact of modality fine-tuning, demonstrating both its benefits and drawbacks on the textual modality, we now explore a potential path towards omni-modal language models. Specifically, we ask: *Is it possible to preserve the positive effects and extend multimodal capabilities without further training the existing models?*

A promising and lightweight approach to addressing this question is *model merging*, which involves integrating the parameters of models with different training corpus and paradigms but the same architecture. Model merging has been shown to be effective in various contexts, including knowledge editing (Lu et al., 2025) and cross-modal knowledge transfer (Ahmed et al., 2022). In the following sections, we explore different model merging strategies and evaluate their effectiveness in preserving textual capabilities while enabling multimodal capability.

### 5.1 MERGING METHODS

We employ two widely used model merging techniques: *average merging* and *weighted average merging*, both of which are task- and modality-agnostic. *Average merging* computes the element-wise average of the weights across all candidate models, while *weighted average merging* assigns a heuristic weight to each model's parameters. Formally, *average merging* is defined as:

$$\theta_{\mathrm{merge}} = \sum_{i=1}^{n} \theta_i,$$ (1)

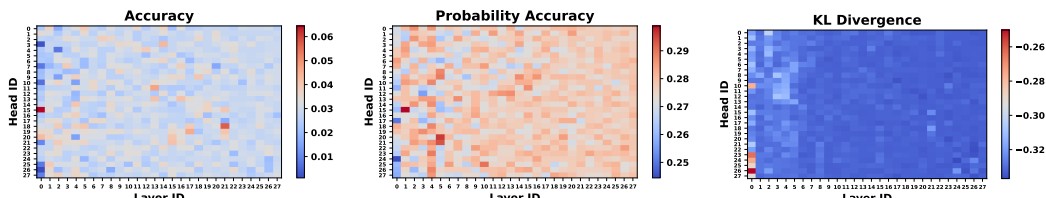

Figure 2: Heat map for masking each attention head. We report accuracy, accuracy calculated by probability, and KL divergence. The accuracy and probability accuracy should be as high as possible, while the KL divergence should have small absolute value. The figure suggests that modality fine-tuning modifies the entire parameter set rather than only specific attention head.

where $\theta_i$ represents the parameters of the $i^{\text{th}}$ candidate model. For *weighted average merging*, the merged parameters are computed as:

$$\theta_{\text{merge}} = \sum_{i=1}^{n} \alpha_i \theta_i, \tag{2}$$

where $\alpha_i$ denotes the weight assigned to the parameters of the $i^{\text{th}}$ model. To ensure the merging process remains both task- and modality-agnostic, the weight $\alpha_i$ is designed to be independent of specific tasks or modalities. This guarantees that newly extended multimodal models can be seamlessly merged without adaptation to modality-specific tasks.

## 5.2 EXAMINATION OF MODEL PARAMETERS

To determine the appropriate design for parameter weights in model merging, we must first answer a fundamental question: *What is the largest unit to which model merging can be applied?* If modality fine-tuning affects only a subset of parameters, merging should ideally be constrained to these altered parameters while preserving the original ones.

To investigate this, we analyze *head-level modality salience*, which quantifies the contribution of individual attention heads to modality-specific tasks. Specifically, we iteratively mask each single attention head and evaluate the resulting impact on model performance, allowing us to assess the relative importance of each head in processing multimodal information. We conduct this analysis using the Qwen2-VL-7B-Instruct model on the MMMU dataset (Yue et al., 2024). We employ three metrics to examine head-level modality salience:

- **Accuracy.** The model is prompted to generate an answer choice directly, and accuracy is computed based on correct predictions. The baseline accuracy of Qwen2-VL-7B-Instruct is 49.44%.
- **Probability Accuracy.** To mitigate the impact of potential degradation in generation quality caused by masking a single attention head, we analyze the logits of the first generated token, expected to correspond to the answer choice. Specifically, we extract the logits of the options (*i.e.*, A, B, C, and D), apply softmax normalization, and compute accuracy.
- **KL Divergence.** To quantify distributional shifts, we compute the Kullback-Leibler (KL) divergence between the option logits of the original model and those after masking.

The results are presented in Figure 2, from which several key observations can be made. Across all three evaluation metrics, **masking any attention head results in a substantial performance drop**, indicating that no single head is dispensable for specific modality processing, unlike retrieval or long context abilities (Wu et al., 2025). This suggests that modality fine-tuning modifies the entire parameter set rather than only specific attention heads, implying that the model merging weight design should account for all parameters rather than a subset of them. Additionally, a notable trend emerges: **attention heads in shallower layers exert a greater influence on multimodal performance.** This observation aligns with the established role of transformer layers, where shallow layers primarily focus on semantic understanding, while deeper layers perform integration and reasoning (Wang et al., 2024a). These findings underscore the importance of preserving early-layer representations when designing model merging strategies for multimodal extensions.

## 5.3 WEIGHTED MODEL MERGING

Since attention heads are too coarse-grained for effective model merging, we refine our approach by considering parameter matrices. To quantify the extent of parameter modifications due to modality

Table 4: Performance of merged LLMs across all evaluated domains of textual abilities. The results show that model merging preserves the most capabilities of base models.

| Model | MMLU Acc | MMLU-Pro Acc | IFEval Avg | PR Acc | ZeroScrolls Acc | GPQA Acc | MATH Acc | HumanEval+ Pass@1 | Pass@5 | Pass@10 | MMMLU Acc | HarmBench ASR@100↓ |
|---|---|---|---|---|---|---|---|---|---|---|---|---|
| Qwen2-7B-Instruct | 66.2 | 35.1 | 58.2 | 100.0 | 85.7 | 13.6 | 60.0 | 54.2 | 82.8 | 93.4 | 51.4 | 15.9 |
| **Average** | | | | | | | | | | | | |
| Qwen2-Text/VL/Video | **68.7** | 31.0 | 58.0 | 100.0 | 81.0 | 10.1 | 55.9 | 49.6 | 83.2 | **94.5** | 52.7 | 18.2 |
| Qwen2-Text/VL/Video/SI | 68.5 | 37.4 | **58.2** | 100.0 | **85.7** | 11.1 | 56.1 | 49.5 | **83.3** | 94.0 | 52.6 | 20.4 |
| Qwen2-VL/Video | **68.7** | 36.0 | 54.8 | 100.0 | **85.7** | 9.6 | 52.9 | 46.0 | 82.2 | 93.2 | 52.5 | 19.4 |
| Qwen2-VL/Vide/SI | 68.4 | **40.0** | 53.6 | 100.0 | **85.7** | 9.6 | 53.9 | 47.8 | 82.9 | 94.2 | 52.3 | 23.7 |
| **Weighted Average** | | | | | | | | | | | | |
| Qwen2-Text/VL/Video/SI | 68.6 | 36.3 | 58.1 | 100.0 | 81.0 | 9.1 | **57.0** | **50.0** | 82.2 | 93.8 | **52.8** | **15.4** |

fine-tuning, we compute $\Delta_{\text{avg}}$ for each tensor, defined as:

$$\Delta_{\text{avg}} = \text{avg}|\theta_{\text{ori}} - \theta_{\text{mft}}|, \qquad (3)$$

where $\theta_{\text{ori}}$ represents the parameters of the original LLM, and $\theta_{\text{mft}}$ denotes those of the modality fine-tuned LLM. This metric captures the average parameter shifts after modality fine-tuning.

Our analysis reveals that Qwen2-VL-7B-Instruct, which undergoes the most extensive modality fine-tuning, exhibits the largest parameter shift from its base LLM, showing 10 times larger parameter shifts than others This observation supports the hypothesis that greater specialization in a modality results in more substantial parameter deviations. Motivated by this insight, we incorporate $\Delta_{\text{avg}}$ into the weight design for model merging. Specifically, for each model parameter $\theta_i$, we first compute $\Delta_{\text{avg}}^i$ using Equation (3). We then apply softmax to the set $\{\Delta_{\text{avg}}^1, \Delta_{\text{avg}}^2, ..., \Delta_{\text{avg}}^n\}$, transforming the values into a probability distribution $\{\alpha_1, \alpha_2, ..., \alpha_n\}$. To preserve the capabilities of the original LLM, we introduce a manually assigned weight $\alpha_0$ for its parameters. The remaining weights are rescaled by multiplying each $\alpha_i$ by $1 - \alpha_0$, ensuring a controlled balance between the original and fine-tuned models. The final weighted-averaged parameter is thus formulated as:

$$\theta_{\text{merge}} = \alpha_0\theta_0 + (1 - \alpha_0)\sum_{i=1}^{n}\alpha_i\theta_i. \qquad (4)$$

## 5.4 RESULTS

We experiment model merging on the Qwen2-7B-Instruct based models, *i.e.*, Qwen2-VL-7B-Instruct, LLaVA-Video-7B-Qwen2, and LLaVA-OneVision-Qwen2-7B-SI. For the evaluation the original textual modality, we follow the setup from Section 4. For the evaluation on other modalities, we adopt the image and video dataset, using MMMU (Yue et al., 2024) and Video-MME (Fu et al., 2024). For generation configuration, we follow the same setup in Section 3.

The results of the textual evaluation are presented in Table 4, while the multimodal evaluation results are shown in Table 5. From these, we derive several key observations.

**Model merging preserves the capabilities of base models.** The textual evaluation results indicate that the merged model retains most of the original LLM's capabilities, with improvements in certain domains.

Table 5: Performance of OLMs based on merged LLMs across all evaluated multimodal domains.

| Model | MMMU | Video-MME |
|---|---|---|
| Qwen2-VL-7B-Instruct | 49.44 | 62.84 |
| Qwen2-avg-all | 48.78 | 56.89 |
| Qwen2-weighted-all | 48.11 | 61.04 |

- **Knowledge**: Modality fine-tuning has been shown to expand the model's knowledge base. Notably, the merged model outperforms even the fine-tuned models, suggesting an enhanced integration of multimodal knowledge.
- **Instruction Following**: While fine-tuned models exhibit a decline in instruction-following ability, merging with the original LLM not only restores but also slightly improves this capability.
- **Long Context**: The merged model maintains performance comparable to the original LLM, indicating that model merging does not degrade this ability.
- **Reasoning**: A consistent performance drop is observed across reasoning tasks following fine-tuning. However, model merging mitigates this decline to some extent.
- **Multilingual**: Performance improves after merging, suggesting that merging helps consolidate multilingual understanding.
- **Safety**: The merged model preserves the safety characteristics of the original LLM.

In summary, except for *reasoning*, the merged model performs on par with or better than the original LLM across evaluated domains. This suggests that future efforts to retain the base LLM's capabilities should focus on addressing reasoning degradation during modality fine-tuning.

**Weighted model merging preserves more model abilities.** Both textual and multimodal evaluations demonstrate that weighted-average model merging achieves more robust performance. This suggests that parameter shift serves as a crucial indicator of a parameter's importance. Furthermore, results indicate that merging a greater number of models further enhances overall performance, highlighting the potential of leveraging multiple modality-extended models to improve omni-modality.

## 6 ON OMNI-MODALITY FINE-TUNING TOWARDS OMNI-MODAL MODEL

Previous sections indicate that model merging still has some degradation in performance. Thus, our next question is: *Is omni-modality fine-tuning the right path towards OLM?* In this section, we will discuss about the effectiveness and the efficiency of omni-modality fine-tuning.

### 6.1 MODALITY FINE-TUNING ON LANGUAGE MODEL

#### 6.1.1 EXPERIMENTAL SETUP

**Model.** For the choice of omni-modality fine-tuned models, we adopt NextGPT (Wu et al., 2024b) and Qwen2.5-Omni (Xu et al., 2025a). The former utilizes a frozen language backbone, while the latter trains the whole model. For the choice of modality fine-tuned models, we choose models that is specialized in certain modality and has the same base LLM as NextGPT, including Instruct-Blip (Dai et al., 2023), LLaVA-Next (Liu et al., 2024), Video-LLaMA (Zhang et al., 2023), Video-LLaVA (Zhang et al., 2024c), and Vista-LLaMA (Ma et al., 2023).

Table 6: Performance of OLMs and modality-specific language models on image and video domains. Red indicates OLMs and blue indicates modality-specific language models.

| Text-Image | | | |
|---|---|---|---|
| Model | Data | VizWiz | VQAv2 |
| NextGPT | 4.5M | 48.40 | 66.70 |
| InstructBlip | 10M | 34.50 | 43.30 |
| LLaVA-Next | 1.3M | 57.60 | 81.80 |

| Text-Video | | | |
|---|---|---|---|
| Model | Data | MSVD-QA | MSRVTT-QA |
| NextGPT | 2.1M | 64.50 | 61.40 |
| Video-LLaMA | 2.8M | 51.60 | - |
| Video-LLaVA | 2M | 70.70 | 59.20 |
| Vista-LLaMA | 1.3M | 65.30 | 60.50 |

**Datasets.** For easier comparison, we adopt the datasets that are used to evaluate NextGPT. For image modality, we adopt VizWiz (Bigham et al., 2010) and VQAv2 (Goyal et al., 2017). For video modality, we adopt MSVD-QA and MSRVTT-QA (Xu et al., 2017).

#### 6.1.2 RESULTS

Our experimental results are presented in Table 6, detailing the training data volumes and model performance across evaluation datasets. We also compare the language capabilities of our merged and fine-tuned omni-modal models in Appendix A.1. The findings reveal that omni-modal fine-tuning is currently less effective and efficient than modality-specialized models.

For image-based tasks, LLaVA-Next requires only one-third of the training data used by NextGPT yet significantly outperforms it on visual understanding benchmarks. Similarly, for video-based tasks, Vista-LLaMA achieves comparable performance to NextGPT while consuming only half the training data. These results suggest that while omni-modality fine-tuning serves as a proof-of-concept for generalizing across modalities, it requires a more refined design to achieve efficiency and performance on par with specialized models. Further research is needed to optimize omni-modality fine-tuning strategies, ensuring they can effectively balance generalization and efficiency without excessive data consumption.

### 6.2 MODALITY FINE-TUNING ON MERGED MODEL

Given that model merging alone is not consistently effective in extending language models to multiple modalities while simultaneously maintaining their core language proficiencies, we shift our focus to employing small-step fine-tuning on the merged model. Previous work (Cohere et al., 2025)

has demonstrated that fine-tuning merged language models with a small number of training steps can enhance their performance across various language-centric abilities. Consequently, we explore whether this conclusion still stands for the multimodal situation. For the base model to conduct fine-tuning on, we utilize the weighted-average merged model detailed in Section 5, *i.e.*, Qwen2-weighted-all. The fine-tuning dataset comprises selections for distinct modalities: MetaMath (Yu et al., 2023) for language, VisualWebInstruct (Jia et al., 2025) for image, and LLaVA-Video-178K (Zhang et al., 2024c) for video. These datasets are chosen because: 1) they are curated for tasks requiring complex reasoning within their specific modality, and 2) they offer a standardized fine-tuning format, facilitating reproducible research. To approximate a balanced token exposure across modalities, we set the data proportion for fine-tuning as text : image : video $= 3 : 2 : 1$.

### 6.2.1 ANALYSIS

**Effects of Fine-Tuning Step Number.** To ground our analysis, we first seek to identify an optimal range for the number of fine-tuning steps. We define optimal small-step fine-tuning as a regimen that 1) enhances modality alignment and performance on multimodal tasks, while 2) not substantially degrading the model's original language capabilities. Performance is evaluated on the MMLU and MMLU-Pro benchmarks and the MMMU benchmark.

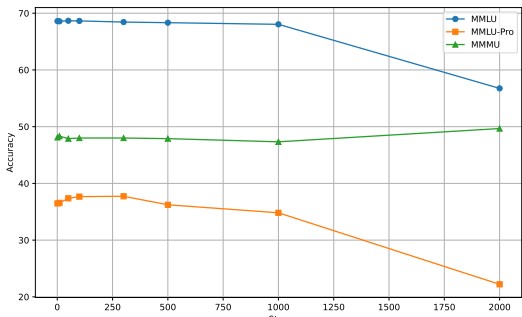

Figure 3: Accuracy over training.

The results, depicted in Figure 3, reveal a critical trend. We observe that performance on both MMLU and MMLU-Pro tends to decrease after approximately 1,000 fine-tuning steps. In contrast, performance on MMMU generally shows improvements with fine-tuning. This divergence strongly suggests a *modality trade-off*: enhancing multimodal understanding through fine-tuning can come at the cost of textual understanding. This observation implies that straightforward small-step fine-tuning may present challenges for developing truly omni-modal models that excel universally across all modalities. Furthermore, textual understanding (MMLU/MMLU-Pro) exhibits a slight increase or peak performance within the initial 100 steps, suggesting that the optimal fine-tuning step number for preserving or enhancing language abilities is relatively small. Conversely, visual and multimodal capabilities (MMMU) may benefit from more fine-tuning steps. This disparity in optimal fine-tuning step number for different modalities likely contributes to the observed trade-off.

Moreover, to examine the impact of modality fine-tuning and model merging on model parameters, we visualize the parameter shift of these models. The results are presented in Appendix A.2.

## 7 CONCLUSION

This work explore the impact of modality fine-tuning on LLMs and evaluated two alternative approaches for developing Omni-Modality Language Models (OLMs): model merging and omni-modality fine-tuning. Modality fine-tuning effectively extends the capabilities of a base LLM to handle multimodal inputs but inevitably alters its parameters. This modification can lead to both improvements in certain domains, such as knowledge expansion, and degradations in core abilities like reasoning and instruction following. Weighted model merging mitigates some of these losses but does not fully preserve all capabilities. Omni-modality fine-tuning, though conceptually promising, proves inefficient compared to modality-specialized models, requiring more training data while offering limited improvements. Overall, our findings suggest that neither modality fine-tuning nor naive omni-modality fine-tuning offers a definitive solution to achieving robust OLMs. We hope this study provides valuable insights for advancing research in multimodal LLMs and inspires new approaches toward achieving truly omni-modality models.

### ETHICAL STATEMENT

This paper presents a study of predominant modality extension methods for LLMs. Our research aims to transparently evaluate the strengths and weaknesses of current techniques to guide future

development toward more robust and reliable models and introduces a weighted-average model merging method. There is no major concern about ethics.

## REPRODUCIBILITY STATEMENT

To ensure the reproducibility of our findings, we will make our implementation publicly available. The complete source code, including scripts for data preprocessing, model training, and evaluation, is provided in the supplementary materials. A detailed description of our experimental setup, including all hyperparameters, is documented in the paper. All datasets employed in this work are publicly available.

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

Table 7: Performance of the merged model compared to the fine-tuned model on language benchmarks. *Abs.* stands for the absolute difference and *Rel.* stands for relative difference.

| | Omni-Modality Fine-tuning | | | | Weighted Merging | | | |
|---|---|---|---|---|---|---|---|---|
| | Qwen2.5-7B | Qwen2.5-Omni-7B | Abs. | Rel. | Qwen2-7B-Inst. | Qwen2-7B-weighted | Abs. | Rel. |
| MMLU-Pro | 56.3 | 47.0 | -9.3 | -16.5% | 35.1 | 36.3 | +1.2 | +3.4% |
| GPQA | 36.4 | 30.8 | -5.6 | -15.4% | 13.6 | 9.1 | -4.5 | -33.1% |
| MATH | 75.5 | 71.5 | -4.0 | -5.3% | 60.0 | 57.0 | -3.0 | -5.0% |
| HumanEval | 84.8 | 78.7 | -6.1 | -7.2% | 54.2 | 50.0 | -4.2 | -7.7% |

# A EXPERIMENTS

## A.1 MODEL MERGING V.S. OMNI-MODALITY FINE-TUNING

Furthermore, both model merging and omni-modal fine-tuning tend to degrade the original language capabilities, demonstrating a decline across most language abilities, particularly in reasoning tasks. However, the merged model shows a slight improvement in language understanding and knowledge-related capabilities. The averaged performance drop for the fine-tuned model is -6.3%, compared to -2.6% for the merged model. This indicates that while both methods impact language skills, the fine-tuning approach appears to be more detrimental to core language abilities than model merging.

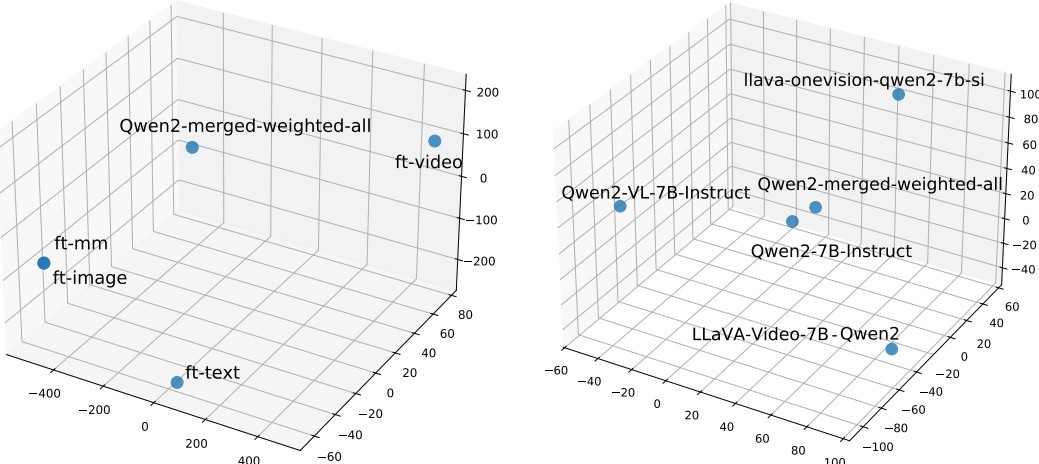

(a) Visualization of weight shifts after fine-tuning on text, image, video, and mixed modality datasets.

(b) Visualization of relative positions of the base models and the weighted-average merged model.

Figure 4: Comparative visualizations of model weight distributions. We use t-SNE to visualize the weight shifts.

## A.2 EFFECTS OF FINE-TUNING V.S. MERGING

To further understand the mechanisms contributing to the *modality trade-off*, we investigate the shifts in model weights induced by the merging process itself versus subsequent fine-tuning. For the fine-tuning aspect of this experiment, we sample 1,000 instances from each modality-specific dataset (text, image, and video) and fine-tune the merged model (*Qwen2-weighted-all*) on these individual sets, as well as on a combined mixed-modality set comprising all 3,000 samples.

The t-SNE (Van der Maaten & Hinton, 2008) visualizations of the weight distributions are presented in Figure 4. From Figure 4a, it is evident that fine-tuning on different modalities propels the model weights in distinct directions within the parameter space. This suggests that fine-tuning encourages specialization towards the statistical properties of the specific modality it is trained on. Conversely, Figure 4b indicates that weighted model merging positions the resultant model in a region that aggregates the weights of the base models.

This fundamental distinction in how weights are manipulated–fine-tuning driving towards special-ized, often divergent points in the weight space versus merging seeking a consensual, interpolated representation–offers a compelling explanation for the observed modality trade-off. While fine-tuning can significantly enhance performance for a target modality, it risks pulling the model's ca-pabilities away from others. Model merging, on the other hand, achieves an initial balance but may not unlock peak performance for any single modality. Subsequent fine-tuning of this merged model, as shown, tends to quickly re-specialize the model, often reintroducing the trade-off by favoring improvement in one area at the expense of another.

## B    THE USE OF LARGE LANGUAGE MODELS

In this manuscript, LLM is utilized as a general-purpose writing assistant. Its role is strictly limited to improving the clarity, conciseness, and grammatical correctness of the text. The LLM is used for tasks such as rephrasing sentences, shortening paragraphs, and polishing the overall prose to meet academic standards. All intellectual contributions, including the research ideation, experimental design, data analysis, and the core arguments presented, are entirely the work of the human authors.

