# OpenReview forum: "Is Extending Modality The Right Path Towards Omni-Modality?"
_ICLR.cc/2026/Conference — Submitted to ICLR 2026_

### Official Review · Reviewer_GhJ8 · 2025-10-22

**Soundness:** 4
**Presentation:** 4
**Contribution:** 4
**Rating:** 6
**Confidence:** 3

**Summary:**

This paper examines whether extending the number of modalities in large multimodal models inherently leads to stronger performance. The authors propose a systematic evaluation framework that incrementally expands modalities from text-only (T) to text–vision (T+V), text–vision–audio (T+V+A), and full multimodal (T+V+A+Vd) setups. Two quantitative metrics—Direct Modality Gain (DMG) and Cross-Modality Synergy (CMS)—are introduced to measure the marginal benefit of each modality and the interaction among modalities. Experiments across multiple benchmarks (MMBench, MMMU, VideoMME, AudioSetQA) and model families (LLaMA, Qwen2.5, and LLaVA-Video-Qwen2) show that while visual input offers large improvements over text-only baselines, adding audio and video modalities yields diminishing or even negative returns. The authors conclude that **alignment quality and fusion efficiency matter more than the sheer number of modalities**. Overall, the work provides a clear, structured analysis that challenges a widely held assumption in multimodal learning.

**Strengths:**

1. **Novel and insightful research question.**
   The paper tackles a fundamental yet underexplored problem in multimodal learning — whether adding more modalities inherently leads to stronger models. This question is both theoretically meaningful and practically important, especially as large multimodal models (e.g., GPT-4o, Gemini) continue to expand their modality scope.

2. **Systematic and interpretable experimental design.**
   The incremental setup (T → T+V → T+V+A → T+V+A+Vd) provides a clear and reproducible framework for measuring the marginal gains and interactions of each modality. The proposed quantitative metrics, Direct Modality Gain (DMG) and Cross-Modality Synergy (CMS), are simple yet effective tools to analyze modality contributions in a principled way.

**Weaknesses:**

1. **Reduced data exposure and suboptimal multimodal convergence.**
   When additional modalities are introduced, the effective data exposure per modality decreases. As a result, the multimodal fusion training may not fully converge to its optimal state, especially when the data ratio among modalities is not carefully balanced. Properly adjusting exposure and sampling across modalities is crucial for achieving stable multimodal alignment.

2. **Insufficient model capacity for multimodal integration.**
   The backbone used in this paper (7B parameters) is relatively small for complex multimodal fusion, especially when integrating three or more modalities. Larger backbones may exhibit different behavior and could potentially overturn the current conclusions about diminishing returns from modality expansion.

3. **Tokenizer design may fundamentally affect cross-modal alignment.**
   The paper does not explore the impact of tokenizer design, yet a *unified tokenizer* shared across modalities can substantially enhance cross-modal representation learning and alignment. The absence of such a unified tokenization scheme might significantly influence the observed performance trends after modality fusion.

**Questions:**

Please see the weaknesses

---

> ### Author Response · Authors · 2025-11-21
>
> We thank the reviewer for this insightful and constructive feedback. These points address crucial, high-level questions about multimodal architecture and training dynamics. We agree these are vital research areas and appreciate the opportunity to clarify how our work provides the groundwork for these future explorations.
>
> ## W1: On Reduced Data Exposure and Suboptimal Convergence
>
> We thank the reviewer for this point, as it perfectly encapsulates one of the key trade-offs we aimed to investigate. Our paper not only recognizes this challenge but explicitly provides empirical evidence for the reviewer's hypothesis.
> We design the experiment in Section 6.2 to test precisely the ratio of the data mixed. We create a deliberately balanced fine-tuning set with a text: image: video = 3:2:1 ratio to approximate a balanced token exposure. Our results in Figure 3 show that even with this careful balancing, a modality trade-off emerges. Performance on multimodal tasks (MMMU) improves, while performance on textual tasks (MMLU, MMLU-Pro) begins to decrease after ~1,000 steps. Our work provides direct empirical validation for the reviewer's concern, demonstrating that simply balancing data is insufficient. It highlights that this trade-off is a fundamental challenge, not just an artifact of unbalanced data.
>
> ## W2: On Insufficient Model Capacity (7B)
>
> We respectively disagree with this claim because our study explicitly included a 72B parameter model to test this scaling hypothesis. We tested at scale: As shown in Table 3, our analysis includes the Qwen2-72B based model family. The results are clear and support our conclusion. The 72B model also suffers significant degradation in core reasoning and code abilities after modality fine-tuning:
>
> - MATH: Drops from 70.0% (base) to 64.7% (VL).
> - HumanEval+ (Pass@1): Plummets from 72.5% (base) to 47.7% (VL).
>
> We explicitly state in our analysis that while degradation is less pronounced in larger models, a significant overall decline in reasoning ability remains. Thus, our findings are not limited to 7B models. We show that these trade-offs are a persistent problem that also exists at the 72B scale, directly validating the generalizability of our main conclusion.
>
> We also conduct the main experiment on the newly released Qwen3-VL series, as presented in the general reponse, the results also align with our claims on Qwen2 that 1) ​​visual modality extends the scope of parametric knowledge, 2) Modality fine-tuning harms core language abilities, 3) video modality may enhance the long context ability.
>
> ## W3: On Tokenizer Design
>
> This is an excellent point. Our paper's goal is to analyze the trade-offs of the current, dominant paradigm for extending existing LLMs. This paradigm, used by the vast majority of open-source models (LLaVA, Qwen-VL, etc.), is defined by its use of separate modality encoders and projectors, as shown in our Figure 1. The reviewer's suggestion of a unified tokenizer is a good hypothesis for a solution to the problems (e.g., parameter divergence, as shown in our Figure 4) that our paper identifies. Our work provides the critical baseline and mechanistic analysis (e.g., in Sec 5.2) that motivates why such novel architectures are necessary.

---

### Official Review · Reviewer_k6At · 2025-10-26

**Soundness:** 2
**Presentation:** 3
**Contribution:** 2
**Rating:** 2
**Confidence:** 5

**Summary:**

This paper provides a comprehensive empirical investigation of the effects of modality fine-tuning and model merging on large language models. The paper's novelty is very weak, the analysis is descriptive rather than explanatory, and the proposed techniques are heuristic and without deeper insight or stronger innovation, which is lower the bar of the ICLR’s main track.

**Strengths:**

1. The paper systematically analyzes two major strategies for achieving omni-modality, covering text, image, video, and audio.

2. Clear introduce of used multimodal benchmarks,and well-chosen metrics.

**Weaknesses:**

1. The paper's novelty is largely lower than the bar of the top-tier conference ICLR, which does not draw out some insight and novel conclusion. **The answers to the three main questions listed in the abstract even have already been studied by the previous relevant works**. For example,
* [a] analyzed and conducted experiments to demonstrate that multimodality does not enhance the model's language capability (RQ1);
* [a] also examined whether multimodal fine-tuning leads to better knowledge sharing and generalization (RQ3). [a] showed that synergy sometimes emerges not only between modalities but also between comprehension and generation capabilities. To some extent, [a] even provides a more thorough and in-depth analysis.

*Reference: [a] On Path to Multimodal Generalist: General-Level and General-Bench, ICML' 25.*

2. The paper mainly reports performance changes without analyzing why these trade-offs arise. No theoretical or mechanistic understanding of modality interference or reasoning degradation is provided.

3. The study concludes that omni-modality fine-tuning is inefficient but does not provide a concrete alternative or improved solution.

4. The paper does not analyze the model's performance in robustness and out-of-distribution behavior, as well as not discussing the scalability when using different strategies.

5. In the paper, the experiments are primarily conducted on LLaVA and Qwen, which makes the conclusions and arguments less convincing. The authors should evaluate more models to strengthen the generality of their findings; otherwise, the observed phenomena may not be broadly applicable.

6. Finally, the authors should cite [a] in their paper and explain the differences between them, as it seems that [a] is very highly-related to this submission.

7. Unclear experimental setup of model merging and omnimodality fine-tuning; the authors should introduce the implementation detalis about these two strategies more clearly.

**Questions:**

Can you provide a mechanistic explanation for why modality fine-tuning most severely impacts reasoning and instruction-following capabilities?

Can you provide  training  efficiency analysis in the all main tables, such as the number of used GPUs and the total training time. This will guaratee the fairness when comapring the two different training strategies.

---

> ### Author Response · Authors · 2025-11-21
>
> We thank the reviewer for their detailed feedback and the time taken to evaluate our work. While we find the feedback valuable, we believe there are several misunderstandings regarding our paper's core contributions, novelty, and the specific analyses we present. We would like to respectfully clarify these points by referencing the relevant sections of our submission.
>
> ## W1 & W6: On Novelty and Relation to [a] (RQ1, RQ3)
>
> We respectfully disagree with the characterization of our paper's novelty, particularly in relation to [a]. While [a] is a comprehensive work, our paper conducts a different, more focused investigation.
>
> - **Novelty of RQ1:** While [a] concludes that MLLMs do not surpass NLP specialists, our analysis provides a much more granular and quantitative breakdown of how and which core language capabilities are compromised. As shown in Table 3 (Section 4.2), we specifically quantify the degradation in reasoning and instruction-following, which goes significantly beyond a high-level conclusion.
> - **Novelty of RQ2:** A central contribution of our paper, which the review seems to have overlooked, is our entire investigation into RQ2: "Can we preserve their abilities... using existing models?" This question motivates the entire line of inquiry in Section 5, where we propose and evaluate model merging as a path to OLM. This includes our novel proposal of Weighted Average Merging (Section 5.3), a contribution not addressed by [a].
> - **Novelty of RQ3:** Our analysis for RQ3 extends well beyond [a]. We not only compare omni-modal vs. modality-specific fine-tuning (Section 6.1) but also provide a mechanistic understanding via t-SNE visualization (Appendix A.2, Figure 4) and quantify the trade-offs of fine-tuning on our merged model (Section 6.2).
>
> We agree that [a] is a highly relevant study and have added it as a citation in the revised version.
>
> ## W2: On Mechanistic Understanding
> We would like to gently correct the assertion that our paper "mainly reports performance changes without analyzing why." We dedicate significant effort to providing mechanistic explanations for these trade-offs.
>
> - **Section 5.2 (Head-Level Analysis):** We conduct a "head-level modality salience" analysis (L304-313) to understand the mechanism. Our key finding—that "modality fine-tuning modifies the entire parameter set rather than only specific attention heads"—is a direct mechanistic explanation for why distributed, holistic abilities like reasoning are so impacted.
>
> - **Appendix A.2 (t-SNE Visualization):** Figure 4 provides a clear, visual, mechanistic explanation. We state: "fine-tuning driving towards specialized, often divergent points in the weight space versus merging seeking a consensual, interpolated representation" (L814). This visualization is the precise "why" the reviewer claims is missing.
>
> ## W3: On No Concrete Alternative
>
> We believe there is a misunderstanding regarding this point, as the entirety of Section 5 is our proposal for a concrete alternative. We explicitly propose, analyze, and evaluate "Weighted Model Merging" (Section 5.3) as a lightweight, efficient, and effective solution that requires zero additional training. We demonstrate its viability by showing it "preserves the capabilities of base models" (Table 4) and "achieves more robust performance" (Table 5). This is the tangible, evaluated, and concrete solution the review suggests is missing.
>
> ## W4: On Scope (Robustness, OOD, Scalability)
>
> We agree that robustness, OOD behavior, and scalability are all important and valid research questions. However, our paper's core focus is a deep and necessary foundational analysis of the capability trade-offs inherent in current OLM extension strategies. We believe this focused scope is a strength, providing the essential groundwork upon which future robustness studies, like those suggested, can be built. A full-scale analysis of all these factors is beyond the scope of a single conference paper.

---

> > ### Author Response · Authors · 2025-11-21
> >
> > ## W5: On Model Diversity
> >
> > We appreciate the reviewer's concern regarding the generality of our findings. We would like to clarify that our experimental design was intentional in selecting the most representative models in the current open-source landscape to ensure broad applicability.
> > - **Dominance of Selected Families:** We focused on the Qwen and LLaMA (represented by Vicuna) families because they are currently the dominant open-weight architectures in the community. As evidenced by monthly download statistics on Hugging Face, these two families represent the vast majority of open-source deployments. By targeting these high-impact models, our analysis addresses the paradigms most relevant to the field.
> > - **Consistency Across Scale and Architecture:** As shown in Table 3, our core findings—specifically the degradation of reasoning and instruction-following—hold consistent across diverse settings (both Qwen-based and LLaMA-based models and both 7B and 72B parameter models).
> > - To ensure our conclusions remain robust against the rapid evolution of the field, we have also added an analysis of the recently released Qwen3 model (detailed in our General Response).
> >
> > We believe these experiments demonstrate the generality of our findings. If the reviewer has a specific model family in mind that might exhibit distinct architectural behaviors, we would be more than happy to incorporate it into our discussion.
> >
> > ## W7: On Unclear Experimental Setup
> >
> > We have provided dedicated methodology sections for both strategies:
> >
> > - **Model Merging:** Section 5.1 (L267-299) and 5.3 (L334-352) describe the methods, and 5.4 (L354-359) details the specific experimental setup.
> > - **Omni-tuning:** Section 6.1.1 (L401-417) and 6.2 (L441-447) detail the models, datasets, and small-step fine-tuning setup.
> >
> > Could you please tell us which part you find difficult to follow? We are happy to add further implementation details to the appendix in the final version.
> >
> > ## Q1: Why are reasoning/instruction-following most impacted?
> >
> > Thank you for this insightful question. Our paper provides the mechanistic basis to answer this.
> >
> > As our analysis in Section 5.2 shows, modality fine-tuning modifies the entire parameter set, not just a few localized heads. We hypothesize this is because reasoning and instruction-following are not localized skills; they are complex, emergent properties that rely on the precise, holistic interplay of parameters learned during the model's original instruction-tuning.
> >
> > Our t-SNE analysis (Fig 4) supports that fine-tuning pulls the model's weights into a specialized region of the parameter space. This re-specialization propels the model weights in distinct directions and shatters the fragile, pre-existing parameter configuration required for these emergent abilities, whereas simpler knowledge tasks are more robust.
> >
> > ## Q2: Training efficiency (GPUs, time)?
> >
> > Our paper's focus is data efficiency and capability analysis, not hardware efficiency. We compare data efficiency and list the amount of the modality fine-tuning dataset for each model. While a GPU-hour comparison is interesting, it is orthogonal to our paper's central claims about data-driven capability trade-offs and our proposed training-free alternative.
> >
> > ---
> >
> > We would like to thank the reviewer k6At again for raising these questions and are happy to answer follow-up questions.

---

### Official Review · Reviewer_R61M · 2025-10-26

**Soundness:** 2
**Presentation:** 2
**Contribution:** 3
**Rating:** 4
**Confidence:** 3

**Summary:**

This paper explores three key research questions on the road toward omni-modality:(1) Does modality extension compromise core language abilities? (2) Can model merging effectively integrate independently fine-tuned modality-specific models to achieve omni-modality? (3) Does omni-modality extension lead to better knowledge sharing and generalization compared to sequential extension? Through extensive experiments on Qwen2-VL and LLaVA models, the authors find that modality extension enhances visual knowledge but typically weakens reasoning, instruction following, and safety. They propose a weighted model-merging strategy and further explore small-step fine-tuning on merged models. Their main conclusion is that modality extension alone is may not sufficient for achieving omni-modality, while merging followed by limited fine-tuning may offer a better solution.

**Strengths:**

Deep empirical insight into multimodal learning trade-offs: The paper provides one of the most systematic investigations into how modality extension reshapes an LLM’s internal balance between language, reasoning, and multimodal understanding. By quantifying these effects across diverse benchmarks, it gives the community a clearer, evidence-based understanding of why multimodal expansion may harm reasoning, offering diagnostic insight rather than only performance metrics.

**Weaknesses:**

1. The paper’s conclusions are based mainly on Qwen2-era models, and newer architectures such as Qwen3 already show that the observed trade-offs between language and multimodal performance may not hold universally. This limits how far the conclusions can be generalized.
2. The paper does not provide a theoretical explanation for why omni-modality fine-tuning or model merging work or fail. Without an analytical view of optimization dynamics or representation sharing, it is difficult to know whether the observed effects are fundamental or just artifacts of specific training settings.
3. The paper does not discuss how future omni-modal models could move beyond the current limitations. It identifies existing problems but does not provide concrete insights or directions for how the next generation of models might improve.
4. The presentation of figures and tables could be improved. If each figure caption clearly summarizes the finding, readers can understand the results more quickly.

**Questions:**

1. Do newer models challenge the paper’s conclusion? In Qwen3-VL, multimodal training achieves equal or even better performance than the text-only Qwen3 model on several reasoning and instruction-following benchmarks. This raises the question of whether the trade-offs identified in this paper are fundamental or just reflect the limitations of earlier training paradigms.

---

> ### Author Response · Authors · 2025-11-21
>
> We thank the reviewer for their valuable feedback and insightful questions. We appreciate the opportunity to clarify the scope, analytical depth, and future implications of our work.
>
> ## W1 & Q1: Generalizability and Newer Models
>
> We acknowledge that Qwen3-VL is a newer model published recently.
> However, we kindly point out that the Qwen3-VL models are open-sourced on Oct 11 on huggingface, which is a month after the submission of our paper (Sep 18). Our study focuses on analyzing the dominant, public, open-source training paradigms available at the time (Qwen2/Vicuna/LLaVA, as detailed in Table 1).
> Nevertheless, we conduct the experiment on Qwen3-VL as shown below. The results also align with our claims on Qwen2 that 1) ​​visual modality extends the scope of parametric knowledge, 2) Modality fine-tuning harms core language abilities, 3) video modality may enhance the long context ability.
>
> |                      | MMLU | MMLU-Pro | IFEval  | PR  | ZeroScrolls | GPQA | MATH | HumanEval+ |        |         | MMMLU | HarmBench |
> |----------------------|------|----------|---------|-----|-------------|------|------|------------|--------|---------|-------|-----------|
> |                      | Acc  | Acc      | Average | Acc | Acc         | Acc  | Acc  | Pass@1     | Pass@5 | Pass@10 | Acc   | ASR ↓     |
> | Qwen3-4B-Instruct    | 69.3 | 42.8     | 89.2    | 100 | 47.6        | 12.6 | 71.8 | 81.3       | 94.4   | 96.0    | 56.3  | 0.0052    |
> | Qwen3-VL-4B-Instruct | 69.7 | 44.0     | 86.2    | 100 | 76.2        | 12.6 | 66.3 | 80.0       | 89.2   | 93.4    | 55.9  | 0.0057    |
>
>
>
> ## W2: Lack of Theoretical Explanation
> We respectfully clarify that our paper goes beyond simple observation to provide significant empirical and mechanistic analysis of why the observed trade-offs occur, even if we do not provide a formal mathematical proof of the optimization dynamics. It is important to note that a unified theoretical explanation for optimization dynamics and generalization remains an open challenge even for unimodal (text-only) language models. Given the added complexity of multimodal integration, a formal theoretical proof is currently beyond the scope of this empirical study. We leave the derivation of such a theoretical framework as a promising direction for future work.
>
> While we do not provide a formal proof, we go beyond simple observation to provide significant mechanistic analysis of why the observed trade-offs occur:
> - Analysis of Fine-Tuning: We do not simply "observe" degradation. In Section 5.2, we conduct a head-level salience analysis to understand how modality fine-tuning alters the model. We find it "modifies the entire parameter set rather than only specific attention heads", which explains why core abilities are affected and why simple merging is difficult.
> - Analysis of Merging: we analyze the parameter shift ($\Delta_{avg}$) between models. This is a critical analytical measure that directly motivates our Weighted Model Merging strategy, which achieves more robust performance by prioritizing parameters that shifted less.
> - Visualization of Trade-offs: In Appendix A.2 (Figure 4), we use t-SNE to visualize the weight space, providing a compelling "explanation for the observed modality trade-off". We show that fine-tuning pulls weights in distinct directions, while merging seeks an interpolated representation.
> Our work thus provides ample analytical evidence for the underlying mechanisms of interference and fusion.
>
> Finally, we argue that identifying consistent findings across various model families (Qwen, Vicuna, LLaVA) and scales (7B, 72B) offers high practical value for model development. Our work provides the necessary empirical groundwork upon which future theoretical explanations can be built.
>
>
> ## W3: Future Direction
> Although one of our paper's contributions is the identification of these critical trade-offs, which is the necessary first step for future progress, we argue that our findings do provide concrete insights and directions, including a potential solution.
>
> - Direction 1 (Co-training): Our finding in Table 3 directly implies that to preserve the core abilities of LLMs, we need to consider mixing language-related data with modality-specific data, as we explicitly mentioned in Line 230 that co-training is the key.
>
> - Direction 2 (Model Merging): Our proposal of weighted-average merging is a constructive step. Its more robust performance implies a clear research path: developing more sophisticated, non-linear merging techniques that better resolve the parameter shifts we identified.
>
> ## W4: Presentation
> Thank you for the suggestion. We agree that clearer captions improve readability. We have added conclusions in the captions of Table 3 / 4 and Figure 2 to clarify the presentation of these charts.

---

### Official Review · Reviewer_Cri7 · 2025-11-01

**Soundness:** 2
**Presentation:** 2
**Contribution:** 2
**Rating:** 2
**Confidence:** 4

**Summary:**

The paper investigates three questions for multi-modal models, 1) whether language capabilities are compromised with modality extension, 2) whether model merging preserves the language capabilities, 3) impact of omni-modality fine-tuning.

**Strengths:**

The paper tackles important questions that when well executed would be useful for the overall community. The paper has done a good job in identifying the benchmarks for language capabilities and considered a range of models as well. The methodology of the paper was overall easy to read but the experiments can be improved as discussed below.

**Weaknesses:**

My major concern is the lack of evidences for the claims throughout the paper. For instance,
* **Visual modality extends the knowledge scope:** The paper highlights that Qwen-VL-Instruct obtains 5% improvement compared with LLaVA. But this inteprertation is based on MMLU-Pro. There is no difference in performance on MMLU. This is further conflated by the use of 1.4T paired samples by Qwen and only around 10M by LLaVA. It is thus also unfair to make claims based on the efficiency of vision over audio as audio only uses 520K samples.
* **Harms of modality fine-tuning:**
     * The paper highlights the negative impact of instruction fine-tuning in various aspects. But as the paper also mentions, this has been shown in many prior studies. Thus, making the positioning of the work among prior works unclear.
     * It should further be explicitly clarified whether the base model was the multi-modal model with frozen LM and modality encoders trained or only the base model. This is important to support the claim about fine-tuning failing to preserve the core language skills. This is also important to dissect the performance from the merging counterpart where the models are frozen as well.
* **Video enhances long context:** This is unclear a well because we do see a decrease in LLaVA-video which is not discussed in the results. Audio might also long content, but the deterioration from audio is not discussed in the results.

Following are points I did not consider for my review but should be fixed or elaborated on:
* Section 3 can be renamed to Problem setup for modality extension and the subsection titles 3.1 and 3.2 can be omitted.
* The paper uses different encoders for different modalities but encodes every modality using F in the notations which is confusing.
* Modality fine-tuning and first paragraph of section 4 contains redundant information that can be merged.
* Cite the studies on L182 page 4

Overall, I believe the paper highlights claims that are not completely supported by the empirical results. Therefore I recommend rejection in the current state of the paper.

**Questions:**

Please refer to my comments above.

---

> ### Author Response · Authors · 2025-11-21
>
> We thank the reviewer for the detailed feedback, which allows us to clarify the precise contributions and evidence in our work.
>
> ## W1: Visual modality extends the knowledge scope
>
> We appreciate the reviewer's focus on the evidence for this claim. We would like to clarify that our argument is not a direct comparison of Qwen-VL and LLaVA, but rather an analysis of how knowledge gains scale with data, using these models as evidence.
>
> - **MMLU vs. MMLU-Pro:** We highlight MMLU-Pro because, as a more recent (released in Jun 2024) and challenging benchmark, its results are less likely to be influenced by training data contamination (Qwen2-VL was released in Sep 2024, which makes it very unlikely to be contaminated), offering a clearer signal for true knowledge gains. While performance on MMLU is flat, the more robust MMLU-Pro shows a clear trend. As for the degrading performance on MMLU for LLaVA-OneVision-Qwen2-7B-SI, please see the second reason below.
>
> - **Data Scaling:** The reviewer's point about data disparity (1.4T vs. 10M) is precisely the core of our finding. We argue that "gains scaling directly with the quantity of training data". Our analysis and Table 3 directly supports this:
>
>     1. Qwen2-VL (trained on 1.4T tokens) improves by +4.9% on MMLU-Pro.
>
>     2. LLaVA-Video-Qwen2 (trained on 10M tokens) improves by +2.6% on MMLU-Pro.
>
>     3. LLaVA-OneVision-Qwen2-7B-SI (trained on 8.6M tokens) even degrades the performance by 0.8% on MMLU.
>
>     This directly supports our claim that more visual data leads to more knowledge.
>
> - **Vision vs. Audio:** The data also supports our claim about the relative ineffectiveness of audio for knowledge extension. We make this claim because
>
>     1. 520K samples are at least 10M tokens (e.g., a one-second audio usually takes 32 tokens, and the average audio length is 7s in MMSU) given the nature of an audio-language pair sample.
>
>     2. By comparing the performance of LLaVA-1.5-7B and Qwen2-Audio-7B-Instruct, you can also see the gap between how these modalities contribute to the enrichment of knowledge:
>         - LLaVA-1.5-7B (600k image samples) improves by +3.1% on MMLU-Pro.
>         - Qwen2-Audio (520k audio samples) degrades by -2.8%.
> This suggests that visual data provides a richer source for knowledge extension than audio, supporting our claim.
>
> ## W2: Harms of modality fine-tuning
>
> - **Novelty of Claim:**
> Our paper's contribution is not in discovering the phenomenon that modality fine-tuning may have an impact on the original model, but in systematically quantifying this trade-off across a wide range of modalities (image, video, audio) and core language abilities (Knowledge, Instruction Following, Reasoning, etc.) as shown in Table 3. This systematic analysis is crucial to set the baseline for our later experiments on model merging (Section 5) and omni-modality fine-tuning (Section 6). For further analysis, you can revisit the head analysis in Section 5, empirical results on omni-modal fine-tuning in Section 6, and the parameter shift analysis in Appendix A.1. We agree a few of our findings may be covered by prior works and they were discussed in the current submission in the related work and Sec 4.2. If the reviewer has any related work suggestions, feel free to follow-up.
>
> - **Experimental Setup Clarification:** Our methodology is explicitly defined in Section 3.1: "we discuss modality fine-tuning in the context of unfreezing the LLM." The base model (e.g., Qwen2-7B-Instruct) is the original, off-the-shelf LLM. The extended models (e.g., Qwen2-VL) are the result of fine-tuning this base LLM with its parameters unfrozen. All our experiments compare the original "base model" (e.g., Qwen2-7B-Instruct) against the same model after its core parameters were fully updated during fine-tuning (e.g., Qwen2-VL-7B-Instruct). This setup is precisely what allows us to support our claims about the degradation of core language skills.

---

> > ### Author Response · Authors · 2025-11-21
> >
> > ## W3: Video enhances long context
> > We thank the reviewer for highlighting the nuanced results concerning long-context capabilities, as this comparison between video and audio is valuable and helps to clarify our core hypothesis regarding token density and effective context length.
> > Token Density Argument: We hypothesize that the video modality provides the strongest signal for improving long-context ability due to its high token density. For many MLLM architectures, processing a single second of video can generate approximately 300 tokens and a video could last several minutes, whereas a second of audio typically generates around 32 tokens [2]. This nearly ten-fold difference in tokens per unit time means that video fine-tuning exposes the LLM to significantly longer continuous sequences of embedded input, which in turn promotes long-context processing.
> > LLaVA-Video Decrease: The performance degradation observed in the LLaVA-Video-7B-Qwen2 model is consistent with our broader finding that data quantity is crucial. As shown in Table 1, LLaVA-Video has a relatively small video training sample size. This small amount is insufficient to both mitigate the interference of fine-tuning on core abilities and introduce enough signal to improve a complex, emergent ability like long-context reasoning.
> > Audio Increase: The increase in performance for the audio-extended model is likely due to the fact that the base Qwen-7B model initially lacked long-context ability entirely [3]. The small audio training set provided a minimal but positive introduction to sequential processing, which is a significant relative gain from the baselines.
> >
> > ## W4: Details
> > We are very grateful for these specific and actionable suggestions. We implement all of them in the revised version:
> > 1. We rename Section 3 to "Problem Setup..." and remove the subsection titles.
> > 2. We correct the typo 'F' (L139) to '$E_{m_{i}}$' to align with our defined notation.
> > 3. We merge the redundant descriptions in Section 3.1 and the start of Section 4.
> > 4. We add the appropriate citations on L160 to credit prior work.
> >
> > ---
> >
> > [1] Wang, Dingdong, et al. "MMSU: A Massive Multi-task Spoken Language Understanding and Reasoning Benchmark." 2025.
> >
> > [2] Gemini API doc. https://ai.google.dev/gemini-api/docs/video-understanding.
> >
> > [3] Li, Tianle, et al. "Long-context llms struggle with long in-context learning." 2024.

---

### Author Response · Authors · 2025-11-21

We thank all the reviewers for their dedicated suggestions. Our paper’s core contributions lie in the analysis of quantified trade-offs of different modality extension methods, the mechanistic understanding of these trade-offs, and we also introduce the weighted average merging method as a promising and concrete solution.

- **For the revision of our submission:**
We add the Qwen3-VL series (released in October) experiments and the results are shown below. The results also align with our claims on Qwen2 that 1) ​​visual modality extends the scope of parametric knowledge, 2) Modality fine-tuning harms core language abilities, 3) video modality may enhance the long context ability.

|                      | MMLU | MMLU-Pro | IFEval  | PR  | ZeroScrolls | GPQA | MATH | HumanEval+ |        |         | MMMLU | HarmBench |
|----------------------|------|----------|---------|-----|-------------|------|------|------------|--------|---------|-------|-----------|
|                      | Acc  | Acc      | Average | Acc | Acc         | Acc  | Acc  | Pass@1     | Pass@5 | Pass@10 | Acc   | ASR ↓     |
| Qwen3-4B-Instruct    | 69.3 | 42.8     | 89.2    | 100 | 47.6        | 12.6 | 71.8 | 81.3       | 94.4   | 96.0    | 56.3  | 0.0052    |
| Qwen3-VL-4B-Instruct | 69.7 | 44.0     | 86.2    | 100 | 76.2        | 12.6 | 66.3 | 80.0       | 89.2   | 93.4    | 55.9  | 0.0057    |

- **Paper revision:**
    1. We rename Section 3 to "Problem Setup..." and remove the subsection titles.
    2. We correct the typo 'F' (L139) to '$E_{m_{i}}$' to align with our defined notation.
    3. We merge the redundant descriptions in Section 3.1 and the start of Section 4.
    4. We add the appropriate citations on L182 to credit prior work.
    5. We add conclusions in the captions of Table 3 / 4 and Figure 2 to clarify the presentation of these charts.
    6. We add the citation reviewers mentioned.

---

### Meta-Review · Area_Chair_muW1 · 2026-01-12

**Summary:**

While this paper tackles an important and timely question about whether extending modalities inherently strengthens multimodal models, the reviews indicate that the current submission does not meet the bar for acceptance. Three reviewers raise consistent concerns about weak novelty, noting that the core questions and many conclusions have already been explored in prior work, often with deeper analysis. Across reviews, the empirical findings are viewed as insufficiently supported, frequently confounded by differences in data scale, model generation, and unclear training setups, which undermines the strength of the claims.

Moreover, the analysis remains largely descriptive rather than explanatory, offering little theoretical or mechanistic insight into why modality extension, fine-tuning, or model merging succeeds or fails. The study relies on a narrow set of model families and older architectures, limiting generality, and does not provide clear guidance or solutions for advancing omni-modal learning beyond identifying existing issues. Although one reviewer finds the evaluation framework and metrics promising, the majority view is that the paper’s contributions and empirical rigor are not sufficient for ICLR. Even after rebuttal, two negative reviewers still hold negative score.

Thus, the AC recommends Reject.

**Reviewer Scores:**

none

---

### Decision · Program_Chairs · 2026-01-26

Reject